# Preparation of Supported Perovskite Catalyst to Purify Membrane Concentrate of Coal Chemical Wastewater in UV-Catalytic Wet Hydrogen Peroxide Oxidation System

**DOI:** 10.3390/ijerph18094906

**Published:** 2021-05-04

**Authors:** Wenwen Zhang, Zhenxue Liu, Pei Chen, Guangzhen Zhou, Zhiying Liu, Yanhua Xu

**Affiliations:** College of Environmental Science and Engineering, Nanjing Tech University, Nanjing 211800, China; zhangwenwen1113@126.com (W.Z.); liuzhenxue@njtech.edu.cn (Z.L.); 202061202053@njtech.edu.cn (P.C.); 202061202038@njtech.edu.cn (G.Z.)

**Keywords:** membrane concentrate, UV-catalytic wet hydrogen peroxide oxidation, supported perovskite catalyst, biotoxicity testing, biodegradability

## Abstract

The effective treatment of membrane concentrate is a major technical challenge faced by the new coal chemical industry. In this study, a supported perovskite catalyst LaCoO_3_/X was prepared by a sol–impregnation two-step method. The feasibility of the supported perovskite catalyst LaCoO_3_/X in the UV-catalytic wet hydrogen peroxide oxidation (UV-CWPO) system for the purification of concentrated liquid of coal chemical wastewater was investigated. The effects of catalyst support, calcination temperature, calcination time, and re-use time on catalytic performance were investigated by batch experiments. The catalysts were characterized by X-ray diffraction (XRD), Scanning electron microscopy (SEM), Brunauer–Emmett–Teller (BET), and X-ray photoelectron spectroscopy (XPS). Experimental results showed that the supported perovskite catalyst LaCoO_3_/CeO_2_ prepared using CeO_2_ as support, calcination temperature of 800 °C, and calcination time of 8 h had the best catalytic effect. The catalytic performance of the catalyst remained excellent after seven cycles. The best prepared catalyst was used in UV-CWPO of coal chemical wastewater membrane concentrate. The effects of H_2_O_2_ dosage, reaction temperature, reaction pressure, and catalyst dosage on UV-CWPO were determined. Under the conditions of H_2_O_2_ dosage of 40 mM, reaction temperature of 120 °C, reaction pressure of 0.5 MPa, catalyst dosage of 1 g/L, pH of 3, and reaction time of 60 min, the removal efficiencies of COD, TOC, and UV_254_ were 89.7%, 84.6%, and 98.1%, respectively. Under the optimal operating conditions, the oxidized effluent changed from high toxicity to non-toxicity, the BOD_5_/COD increased from 0.02 to 0.412, and the biodegradability of the oxidized effluent was greatly improved. The catalyst has a simple synthesis procedure, excellent catalytic performance, and great potential in the practical application of coal chemical wastewater treatment.

## 1. Introduction

Membrane separation technology is often used in the advanced treatment unit of coal chemical wastewater to meet the requirement of “zero discharge” in the coal chemical industry [1]. Membrane separation technology can selectively separate impurities in coal gasification wastewater at the molecular or ion level, but cannot degrade pollutants [2], so a certain amount of membrane concentrate will be produced during treatment. Membrane concentrate has the characteristics of high organic concentration, large chroma, and poor biodegradability [3]. The effective treatment of concentrated membrane solution is a major challenge in the advanced treatment of coal chemical wastewater. Ultraviolet-catalytic wet hydrogen peroxide oxidation (UV-CWPO) technology introduces ultraviolet light into catalytic wet hydrogen peroxide oxidation (CWPO) system to combine photocatalysis and CWPO systems [4,5]. UV-CWPO can completely oxidize refractory organic matter in wastewater and intermediate products produced in wet oxidation into carbon dioxide, H_2_O, and biodegradable small molecules; this method can deodorize, decolorize, and disinfect at the same time for harmless treatment of toxic wastewater. In addition, UV light itself can make H_2_O_2_ produce ·OH with strong oxidation, and the synergistic effect of UV light and catalyst can produce more ·OH more efficiently. Compared with CWPO systems, conventional CWPO systems operate at lower temperatures and pressure levels due to the introduction of UV [6].Some studies have shown the superiority of photocatalytic system. Zazo et al. [7] found that UV radiation can promote the reduction of Fe^3+^ to Fe^2+^ on the surface of the catalyst, so that more H_2_O_2_ can be decomposed into hydroxyl radical ·OH. In addition, UV radiation can easily mineralize oxalic acid, which greatly reduces the leaching of active phase and prolongs the service life of the catalyst. The catalyst is the key to UV-CWPO technology. The active components in a catalyst can reduce the reaction conditions and improve the reaction efficiency. Therefore, finding a highly efficient and stable catalyst is an important challenge to popularize this technology [8].

A perovskite-type catalyst is a composite metal oxide with a cubic crystal structure and has the advantages of controllable structure, good thermal stability, high catalytic efficiency, and strong redox ability [9]. This catalyst is widely used in flue gas desulfurization, catalytic combustion, tail gas purification, wastewater treatment, and other fields [10,11,12]. Hsu et al. [13] successfully synthesized LaFeO_3_ by sol–gel method. The removal efficiency of tricholorethylene (TCE) in aqueous solution using LaFeO_3_ (2 g/L) as photocatalyst can reach 95% within 1 h after xenon lamp irradiation. Burcu Palas et al. [14] used LaNiO_3_ perovskite catalyst in the decolorization study of catalytic wet oxidation of active black 5 azo dye wastewater. Under the conditions of catalyst dosage of 1 g/L, reaction temperature of 50 °C, reaction pressure of 1 MPa, pH of 3, and initial solution concentration of 100 mg/L, the degradation and decolorization rates of reactive black 5 solution were 65.4% and 89.6%, respectively. The perovskite-type catalyst has great application prospect in the field of UV-CWPO. It cannot only meet the performance requirements of CWPO system, but also fully cooperate with ultraviolet light in the system. However, perovskite-type catalysts generally have the disadvantage of small specific surface area. Supporting these catalysts with large specific surface area to obtain more dispersed active components is a research hotspot [15].

This study used sol impregnation method and selected a variety of supports to synthesize supported perovskite catalyst. The catalytic activity of several benchmark catalysts was tested to screen a desirable carrier and obtain high-quality supported perovskite catalyst LaCoO_3_/X. The effects of calcination temperature and duration on the wastewater purification efficiency of the catalyst were investigated by batch experiments. The physical and chemical properties of the catalyst were characterized by XRD, SEM, BET, and XPS. Finally, the catalysts obtained under the optimal preparation conditions were used in the UV-CWPO system to purify coal chemical wastewater membrane concentrate. The optimal process conditions including H_2_O_2_ dosage, catalyst dosage, reaction temperature, and reaction pressure were explored. Toxicity test and biodegradability analysis of the oxidized effluent were also carried out.

## 2. Materials and Methods

### 2.1. Chemicals and Materials

All chemicals used in this study were analytical reagents without further purification. Lanthanum nitrate (La(NO_3_)_3_·6H_2_O) was purchased from Shanghai Lingfeng Chemical Reagent Co., Ltd. (Shanghai, China). Cobalt nitrate (Co(NO_3_)_2_·6H_2_O), citric acid (C_6_H_8_O_7_·H_2_O), cerium oxide (CeO_2_), potassium dichromate (K_2_Cr_2_O_7_), and silver sulfate (Ag_2_SO_4_) were acquired from Sinopharm Reagent Co., Ltd. (Shanghai, China). Activated alumina, nano titanium oxide (TiO_2_), concentrated sulfuric acid (H_2_SO_4_), sodium hydroxide (NaOH), hydrogen peroxide (H_2_O_2_, 30%), and phenanthroline were obtained from Nanjing Chemical Reagent Co., Ltd. (Nanjing, China). Mercury sulfate (HgSO_4_) and ammonium ferrous sulfate (H_8_FeN_2_O_8_S_2_·6H_2_O) were provided by Shanghai Triangsihewei Chemical Co., Ltd. (Shanghai, China). Deionized water was produced by a pure water treatment system (EPED-Z1-30T, Eped Technology, Beijing, China) and used to prepare all solutions (resistivity ≥ 1 MΩ·cm).

Wastewater was obtained from a coal-to-natural gas wastewater advanced treatment unit in Inner Mongolia. Membrane concentrate characteristics were COD 1510 mg/L, pH 7.86, TOC 658 mg/L, UV_254_ 1.562, NH4‒N 3170 mg/L, and total dissolved solid content 6200 mg/L.

### 2.2. Preparation of Materials

Supported perovskite catalyst LaCoO_3_/X (X = CeO_2_, TiO_2_, Al_2_O_3_) was prepared by sol impregnation method. According to the stoichiometric ratio (La(NO_3_)_3_·6H_2_O): (Co(NO_3_)_2_·6H_2_O) = 1:1, (metal ion):(C_6_H_8_O_7_·H_2_O) = 1:1.5, certain amounts of La(NO_3_)_3_·6H_2_O, Co(NO_3_)_2_·6H_2_O, and C_6_H_8_O_7_·H_2_O were weighed and dissolved in deionized water to form a mixed solution. Several carriers were impregnated in the solution by constant-volume impregnation method. The mixed solution was oscillated in an ultrasonic oscillator for 30 min, filtered, and placed in an oven. The samples were dried at 105 °C until they were completely dried. The samples were finely ground and placed in a muffle furnace. Perovskite catalyst LaCoO_3_/X was obtained by roasting at the corresponding temperature(500–1000 °C) for the corresponding time (5–10 h).

The crystal phase structure of the catalyst was characterized by X-ray diffraction analyzer (XRD, XD-6 type, Beijing Purkay General Instrument Co. Ltd., Beijing, China). At 36 kV and 20 mA, XRD analysis was carried out under monochromatic Cu Kα radiation (*λ* = 1.54056 Å) within the 2θ scan range of 10°–80°.The specific surface area, pore volume, and pore size distribution of the catalysts were measured using an N_2_ adsorption and desorption analyzer [BET, Micromeritics TriStar Ⅱ type 3020, McMurdo red 2 g (Shanghai, China) Instrument Co., Ltd.]. Scanning electron microscope (SEM, S3400N Ⅱ type, Japan Hitachi) was used to observe the surface morphology and analyze the load cases of the catalysts. Before the test, a thin layer of gold was plated on the sample surface to improve the electrical conductivity under the test voltage of 10.00 kV. The valence states of elements on the catalyst surface were analyzed by X-ray photoelectron spectroscopy (XPS, AXIS-ULTRA DLD-600 W, Shimadzu, Japan). Al Kα was used as excitation source, and the experimental resolution was higher than 0.1 eV. All recorded lines were calibrated to C1s at 284.8 eV line.

### 2.3. Degradation Experiments and Analytical Method

A UV-CWPO system was used to treat coal chemical wastewater membrane concentrate.The UV-CWPO system consists of a 500 mL photochemical high-pressure reactor (TFM-500, (Beijing Zhongjiao Jinyuan Technology Co., Ltd., Beijing, China) and a 30W UV deuterium light source (CE-DD30T, Beijing Zhongjiao Jinyuan Technology Co., Ltd., Beijing, China), as shown in Figure 1.

Three catalytic oxidation experiments were conducted in parallel at the same time, and the average value was obtained for subsequent data analysis and discussion. In an independent catalytic oxidation experiment, 250 mL of water sample was mixed with dilute sulfuric acid to adjust the pH to 3 and placed in a 500 mL photochemical high-pressure reactor. Certain amounts of H_2_O_2_ (10–60 mM) and catalyst (0.4–1.4 g/L) were added, and air at certain pressure (0.5–1.7 MPa) was added. The reaction temperature was set at 60 °C–160 °C, and the 30 W UV-deuterium lamp source was turned on when the set temperature was reached. Samples were obtained at different reaction intervals for analysis. The recycling performance of the catalyst was investigated under the optimal process conditions. The COD of the water samples was determined by the potassium dichromate method. Total organic carbon analyzer (TOC, TOC-L, Shimadzu Company, Osaka, Japan) was used to measure the TOC of water samples. Generally speaking, the UV_254_ value is the absorbance of some organic matter in water under the ultraviolet light at 254 nm. UV_254_ reflects the quantity of humic macromolecular organic compounds and aromatic compounds containing C=C double bond and C=O double bond naturally occurring in water. Phenolic compounds were the main representative compounds in the membrane concentrate, thereby it was necessary to choose UV_254_ as the water quality index. A UV-Vis spectrophotometer (UV-2600, Shimadzu Company, Osaka, Japan) was used to measure water UV_254_ at 254 nm wavelength.

The BOD_5_ of the water samples was determined using hash BOD tester (BODTrak II type, Loveland, CO, USA) to investigate biochemical changes during liquid waste processing. The method for determination of the influence of chemical substances on the microbial metabolism of the Organization for Economic Cooperation and Development (OCED) was used as reference. The influence of the wastewater sample on the respiration rate of activated sludge was measured by a biological respiration meter (PF-8100, RSA Corporation, Danbury, CT, USA) to assess the biological toxicity of wastewater. The pH of wastewater to be tested was adjusted to 7.5 ± 0.5 by adding sodium hydroxide or hydrochloric acid to ensure the acid–base environment required for normal growth of activated sludge. About 250 mL of the activated sludge was placed in a 500 mL reaction flask. OCED medium and water samples were added. After the test was started, the biological respirator automatically recorded oxygen consumed in the reactor. According to the oxygen consumption rate in a certain period of time, respiratory inhibition rate was calculated by Formula (1) [16].
Respiratory depression rate = (OUR_blank_ − OUR_sample_)/OUR_blank_(1)

## 3. Results and Discussion

### 3.1. Optimization of Catalyst Preparation Conditions

#### 3.1.1. Effect of Catalyst Support

Figure 2 shows the influence of different supports on the catalytic effect of the catalyst. The order of catalytic performance is: LaCoO_3_/CeO_2_ > LaCoO_3_/TiO_2_ > LaCoO_3_ > LaCoO_3_/Al_2_O_3_. The supported perovskite catalyst LaCoO_3_/Al_2_O_3_ had the worst catalytic effect. The COD removal efficiency was only 77.5% when the reaction time reached 90 min, which was lower than the COD removal efficiency (80.6%) of LaCoO_3_ in the same reaction time. When Al_2_O_3_ was used as the carrier, Al_2_O_3_ invaded the lattice of LaCoO_3_ and changed the inherent structure of LaCoO_3_, thereby affecting the catalytic activity of the LaCoO_3_/Al_2_O_3_ composite material [17]. The supported perovskite catalysts LaCoO_3_/CeO_2_ and LaCoO_3_/TiO_2_ had better catalytic effects than the original LaCoO_3_, and the former had the best catalytic performance. The COD removal efficiency reached 89.5% when the reaction reached 90 min. The catalytic activity of the supported TiO_2_ catalyst was improved because the support (TiO_2_) produced photocatalytic effect with ultraviolet light in the system [18]. When CeO_2_ was used as the support, CeO_2_ improved the thermal stability and oxygen storage capacity of the catalyst. Cerium-based catalysts would not enter the crystal lattice of perovskite, similar to commonly used transition metal cations (e.g., La, Co, Fe, Mn). Thus, LaCoO_3_/CeO_2_ exhibited better catalytic activity [19]. Therefore, CeO_2_ was selected as catalyst support to prepare supported perovskite catalyst LaCoO_3_/CeO_2_ for subsequent experiments.

#### 3.1.2. Effect of Calcination Temperature

The effects of different calcination temperatures (500 °C, 600 °C, 700 °C, 800 °C, 900 °C, and 1000 °C) on the catalytic activity of LaCoO_3_/CeO_2_ were investigated (Figure 3). With increasing roasting temperature, the catalytic effect of the catalyst was gradually improved.The catalyst with calcination temperature of 800 °C performed the best catalytic effect, and the COD removal efficiency reached 90.2%. The catalytic activity of the catalyst did not continue to increase and even began to decrease when the calcination temperature exceeded 800 °C. The COD removal efficiency only reached 85.6% by the catalyst with the calcination temperature of 1000 °C. The perovskite structure of the catalyst on the CeO_2_ carrier had not been formed when the calcination temperature was in a lower range, and the purity of the effective perovskite structure was also relatively low, resulting in lower catalyst activity [20]. Meanwhile, if the calcination temperature was too high, the catalyst was prone to sintering, resulting in the collapse of the internal structure of the catalyst, blockage of the pores, and reduction of active sites, which in turn led to the degradation of the catalytic activity [21]. In summary, 800 °C was selected as the best calcination temperature of LaCoO_3_/CeO_2_.

#### 3.1.3. Effect of Calcination Time

The effects of different calcination times (5, 6, 7, 8, 9, and 10 h) on the catalytic activity of LaCoO_3_/CeO_2_ were investigated (Figure 4). As the calcination time of the catalyst increased, the catalytic effect gradually improved. When the calcination time was 8 h, the catalytic effect reached the highest, and the COD removal efficiency reached 91.1%. As the calcination time continued to increase, the catalytic activity of the catalyst began to decrease gradually. The COD removal efficiency only reached 86.4% by the catalyst after calcination 10 h. Due to the short calcination process, the complexing agent in the catalyst was not completely burned out, so the structure of the catalyst was unstable, the pore was not clear, and the active point formed was less. Thus, the catalyst with a shorter calcination time showed poor catalytic performance [22].

When the calcination time was too long, a perovskite catalyst structure was formed on the CeO_2_ support and part of the catalyst was sintered and agglomerated, thereby decreasing the specific surface area [23], destroying the active point of the catalyst, and reducing the catalytic effect. Therefore, 8 h was selected as the best roasting time for the catalyst.

### 3.2. Characterization of Catalysts

#### 3.2.1. X-ray Diffraction (XRD)

The crystal phase structures of LaCoO_3_ and LaCoO_3_/CeO_2_ are shown in Figure 5. Characteristic diffraction peaks of LaCoO_3_ perovskite structure prepared by the sol–gel method were found at 33.04°, 47.65°, and 59.08°, consistent with the standard card of LaCoO_3_ (PDF#48-0123). This finding indicated the effectiveness of the preparation method. The prepared catalyst had a pure perovskite structure and orthogonal crystal. The XRD spectra of the supported catalyst LaCoO_3_/CeO_2_ prepared by impregnation showed not only obvious characteristic diffraction peaks of LaCoO_3_ perovskite structure but also characteristic peaks of CeO_2_ at 29.08°, 33.04°, and 56.51°, corresponding to the standard card of CeO_2_ (PDF#1-800) [24]. This result confirmed that an LaCoO_3_ perovskite-type structure was formed on the surface of the CeO_2_ support, and the crystal forms of the support and active component were not destroyed.

#### 3.2.2. N_2_ Adsorption/Desorption Analysis

The specific surface area and pore texture information of LaCoO_3_ and LaCoO_3_/CeO_2_ are listed in Table 1. The specific surface area of LaCoO_3_/CeO_2_ increased from 2.685 m^2^/g of pure LaCoO_3_ to 13.073 m^2^/g, the pore volume increased from 0.006297 cm^3^/g to 0.052408 cm^3^/g, and the pore diameter was 4.0569 nm. With increasing specific surface area and pore volume, the active points on the catalyst surface also increased, and the catalytic activity became stronger [25].

#### 3.2.3. Scanning Electron Microscopy (SEM)

Figure 6 shows the morphology and structure of LaCoO_3_/CeO_2_. As shown in Figure 6a, the composite metal oxide LaCoO_3_ was distributed on the surface of the CeO_2_ carrier in the form of uniform crystal particles. Although particles were closely connected, the gap between them were clearly seen. In addition, although the LaCoO_3_ particles on the surface of the support are partially sintered, there are still a large number of pores on the crystal surface and between the crystal structures (Figure 6b,c). Increasing the active point of the catalyst and the contact area between the catalyst and the reactants was conducive to the catalytic reaction [26]. In conclusion, the supported LaCoO_3_/CeO_2_ catalyst had a larger specific surface area and higher catalytic activity, consistent with the microscopic characterization and macroscopic experimental results.

#### 3.2.4. X-ray Photoelectron Spectroscopy (XPS)

The surface element analysis results of the LaCoO_3_/CeO_2_ catalysts are shown in Figure 7, Figure 8 and Figure 9. Figure 7a,b shows the La 3D XPS spectra of LaCoO_3_ and LaCoO_3_/CeO_2_ catalysts, respectively. Two pairs of bimodals appeared in both spectra. Two pairs of bimodals located at the lower binding energy (A: 833.8 and 837.7 eV; B: 834.2 and 838.1 eV) were attributed to La 3d_3/2_ and La 3d_5/2_. The two pairs of double peaks located at the higher binding energy position were the carrying peak generated by the main peak orbital spin fission, consistent with the characteristic peak of La^3+^ [27] and confirming the existence of La in LaCoO_3_ and LaCoO_3_/CeO_2_ catalysts. Figure 8a,b shows the XPS spectra of Co2P of LaCoO_3_ and LaCoO_3_/CeO_2_ catalysts, respectively, in the vicinity of 795.5 eV (a: 795.1 eV; B: 795.8 eV) and around 780 eV (A: 779.8 eV; B: 780.1 eV) were attributed to Co 2P_1/2_ and Co 2P_3/2_, respectively. Peak-splitting fitting was performed for Co 2P_3/2_, which was near 779 eV (A: 779.1 eV; B: 779.3 eV) and 781 eV (A: 780.9 eV; B: 781.3 eV) corresponding to Co^3+^ and Co^2+^, respectively. This finding suggested that Co in LaCoO_3_ and LaCoO_3_/CeO_2_ catalysts existed in the form of CO^3+^ and CO^2+^, respectively [28].

Figure 9a,b show the XPS spectra of O 1s of LaCoO_3_ and LaCoO_3_/CeO_2_ catalysts, respectively. At around 528.5 eV in the two spectra (A: 528.3 eV; B: 528.7 eV) and 531 eV (A: 530.8 eV; B: 531.2 eV), the two main peaks were attributed to lattice oxygen (O_L_) and surface adsorbed oxygen (O_S_) [29]. The relative contents of lattice oxygen (O_L_) and surface adsorbed oxygen (O_S_) in LaCoO_3_ and LaCoO_3_/CeO_2_ catalysts were calculated according to the ratio of the peak area of Co^3+^ and Co^2+^ (Table 2). With the addition of CeO_2_ support, the ratio of Co^3+^/Co^2+^ in the catalyst was close to 1, indicating that the mutual conversion between Co^3+^ and Co^2+^ in LaCoO_3_/CeO_2_ supported catalyst gradually reached a balance. The oxygen vacancy was formed during charge balance, which was beneficial to the improvement of catalytic activity [30]. The O_L_/O_S_ ratio increased from 0.80 to 1.48, indicating that the lattice oxygen defects increased in the supported catalysts. The lattice oxygen defect was beneficial to increase the active point of the catalyst, thus improving the catalytic effect of the catalyst [31].

### 3.3. Degradation Experiments

#### 3.3.1. Comparison to Several Kinds of Reaction Systems

Figure 10 showed that COD removal efficiency of wastewater by UV photocatalysis was quite low, only 16.6%. The reason for this phenomenon was that UV photocatalytic oxidation was limited by reaction conditions and light could not penetrate the turbid solution. The COD removal efficiency in wastewater by CWPO was 75.9%. The removal efficiency of COD in wastewater by UV-CWPO was the best. COD removal efficiency as high as 89.7% when the reaction reached 60 min, and its trend was basically stable after 60 min. Therefore, the UV-CWPO system proves that the combination of UV and CWPO technologies can produce beneficial effects and improve the COD removal efficiency.

#### 3.3.2. Effect of H_2_O_2_ Dosage on Wastewater Purification

The influence of different H_2_O_2_ dosages on wastewater purification is shown in Figure 11. With increasing H_2_O_2_ dosage, the removal efficiencies of COD, TOC, and UV_254_ in wastewater increased. When the H_2_O_2_ dosage reached 40 mM, the removal efficiencies of COD, TOC, and UV_254_ reached the maximum values of 88.5%, 81.2%, and 97.9%, respectively. When the H_2_O_2_ dosage exceeded 40 mM, the removal efficiency of UV_254_ had no significant increase, while the removal efficiencies of COD and TOC showed a downward trend. With the increase of H_2_O_2_ dosage, more ·OH was produced, which enhanced the oxidation capacity of the reaction system [32]. When the concentration of H_2_O_2_ was too high, excess H_2_O_2_ was decomposed with ·OH in the reaction system, inhibiting the oxidation capacity of the system and causing the waste of oxidants [33]. Overall, the optimal dosage of H_2_O_2_ was 40 mM.

#### 3.3.3. Effect of Reaction Temperature on Wastewater Purification

The effect of different reaction temperatures on wastewater purification is shown in Figure 12. As the reaction temperature increased within 60 °C–160 °C, the removal efficiencies of COD, TOC, and UV_254_ in wastewater increased continuously at first and then tended to be stable. When the reaction temperature was 120 °C, the removal efficiencies of COD, TOC, and UV_254_ reached the maximum values of 88.5%, 86.7%, and 97.9%, respectively. The gradual increase in the reaction temperature in the initial stage accelerated the reaction process between organic matter in wastewater and ·OH, thereby continuously increasing the rate of the reaction system [34]. When the temperature continued to rise, the removal efficiencies of COD, TOC, and UV_254_ tended to be stable without significant changes. Therefore, the optimal reaction temperature was 120 °C.

#### 3.3.4. Effect of Reaction Pressure on Effluent Purification

The effect of different reaction pressure levels on wastewater purification is shown in Figure 13. With increasing reaction pressure, the removal efficiencies of COD, TOC, and UV_254_ remained basically unchanged. When the reaction pressure was 0.5 MPa (the saturated vapor pressure of water at 120 °C is 0.2 MPa), the removal efficiencies of COD, TOC, and UV_254_ reached 89.2%, 84.5%, and 97.1%, respectively. Increase in the reaction pressure had no significant effect on oxidation. In the UV-CWPO system, the oxidation process between H_2_O_2_ and pollutants is the main body, and the mass transfer of oxygen from the gas phase to the liquid phase is less than that in the CWPO system, thereby reducing energy consumption [35,36]. Therefore, the system can oxidize and degrade pollutants under low-pressure conditions. Considering the oxidation effect and safety, the optimal reaction pressure was 0.5 MPa.

#### 3.3.5. Effect of Catalyst Dosage on Wastewater Purification

The effect of different dosages of catalyst on wastewater purification is shown in Figure 14. With increasing catalyst dosage, the removal efficiencies of COD, TOC, and UV_254_ in wastewater showed a trend of increasing first and then decreasing. When the dosage of the catalyst was 1 g/L, the removal efficiencies of COD, TOC, and UV_254_ reached 89.7%, 84.6%, and 98.1%, respectively. With increasing catalyst dosage, the active points on the catalyst surface increased, and the decomposition rate of H_2_O_2_ to produce ·OH was accelerated, thereby increasing the oxidation rate. When the dosage of the catalyst was greater than 1 g/L, the light transmittance of wastewater was negatively affected, which was not conducive to the absorption of ultraviolet light. The oxidation effect then decreased slightly [37]. Therefore, the optimal dosage of the catalyst was 1 g/L.

### 3.4. Reusability and Stability of Catalyst

Seven cyclic degradation experiments were carried out to evaluate the reusability and stability of the LaCoO_3_/CeO_2_ catalyst (Figure 15). The removal efficiencies of COD, TOC, and UV_254_ reached 88.3%, 82.8%, and 96.3% after seven times of repeated use, which were not significantly lower than the removal efficiencies of the first use. Hence, the LaCoO_3_/CeO_2_ catalyst had good reusability and stability.

## 4. Biodegradability Analysis

### 4.1. Biodegradability

Change in biodegradability before and after wastewater purification is very important for actual water treatment. First, BOD_5_/COD before and after wastewater purification in the UV-CWPO system was tested (Figure 16). In general, the greater the BOD5/COD value is, the better the biodegradability of wastewater is. The wastewater is generally considered to be biodegradable when BOD_5/_COD > 0.3. The biodegradability of the concentrated liquid of coal chemical wastewater membrane before oxidation was very poor. The BOD_5_/COD was only 0.02 after five days of biochemical cultivation, indicating that wastewater before oxidation treatment could not be directly treated in the biochemical unit. The biodegradability of wastewater purified by this system was greatly improved. The BOD_5_/COD of the wastewater reached 0.412 after five days of biochemical culture, and the biodegradability was greatly improved. After oxidation, the large molecular organic matter and toxic and harmful components in wastewater were greatly degraded and transformed into small molecular substances that can be biodegraded [38]. Therefore, wastewater oxidized by UV-CWPO can be treated in the biochemical unit.

### 4.2. Biological Toxicity Analysis

The biological toxicity of wastewater before and after treatment was determined by measuring the effect of wastewater on the respiration rate of activated sludge. Figure 17 shows the oxygen consumption rate (OUR) of the sample (oxygen mass consumed per unit time). When the reaction lasted for 8 h, the OUR of blank water sample was 2.02 mg/h, the OUR of raw water was 0.33 mg/h, and the OUR of the effluent was 2.28 mg/h. The OUR of water oxidized by the UV-CWPO system was higher than that of the blank water sample. The OUR of the raw water was lower than that of the blank water sample. This finding indicated that the concentrated liquid of coal chemical wastewater membrane before oxidation had serious inhibitory effect on the respiration rate of activated sludge. However, the oxidized water sample promoted the respiration rate of activated sludge, confirming that the oxidized water sample is easily degraded by activated sludge [39].

The inhibition rates of water samples before and after oxidation treatment were calculated relative to the blank samples, and the toxicity of wastewater was classified accordingly (Table 3). The water sample after oxidation treatment changed from the original highly toxic to non-toxic. Hence, the oxidized effluent can enter into the biochemical unit for treatment.

## 5. Conclusions

In this study, supported perovskite catalyst LaCoO_3_/X was prepared by two steps (sol–gel method and impregnation). The best catalyst LaCoO_3_/CeO_2_ was selected based on macro catalyst performance test and micro characterization analysis. The optimal preparation conditions were established (calcination temperature of 800 °C, calcination time of 8 h). The overall crystal structure of LaCoO_3_/CeO_2_ was clear, and the composite metal oxide LaCoO_3_ with standard perovskite structure was well distributed on the surface of the CeO_2_ carrier. LaCoO_3_/CeO_2_ was used as the catalyst in the UV-CWPO system to participate in the purification of coal chemical wastewater membrane concentrate. Under the best process conditions, i.e., pH of 3, reaction time of 60 min, H_2_O_2_ dosage of 40 mM, reaction temperature of 120 °C, reaction pressure of 0.5 MPa, and catalyst dosage of 1 g/L, the removal efficiency of COD, TOC, and UV_254_ reached 89.7%, 84.6%, and 98.1%, respectively. In addition, the removal efficiency of COD, TOC, and UV_254_ still reached 88.7%, 82.8%, and 96.3% after seven cycles of degradation, which were not significantly lower than those at the first use. Apparently, the prepared LaCoO_3_/CeO_2_ catalyst showed desirable reusability and stability. The biodegradability analysis showed that the BOD_5_/COD of wastewater treated by the UV-CWPO system increased from 0.02 to 0.412. The toxicity analysis indicated that the water sample after oxidation changed from the original highly toxic to non-toxic, and the oxidized effluent was suitable for subsequent biochemical units. Hence, the supported perovskite catalyst LaCoO_3_/CeO_2_ is a pure perovskite catalyst with orthogonal crystal form and has a good application prospect for treating concentrated liquid of a coal chemical wastewater membrane in a UV-CWPO system.

## Figures and Tables

**Figure 1 ijerph-18-04906-f001:**
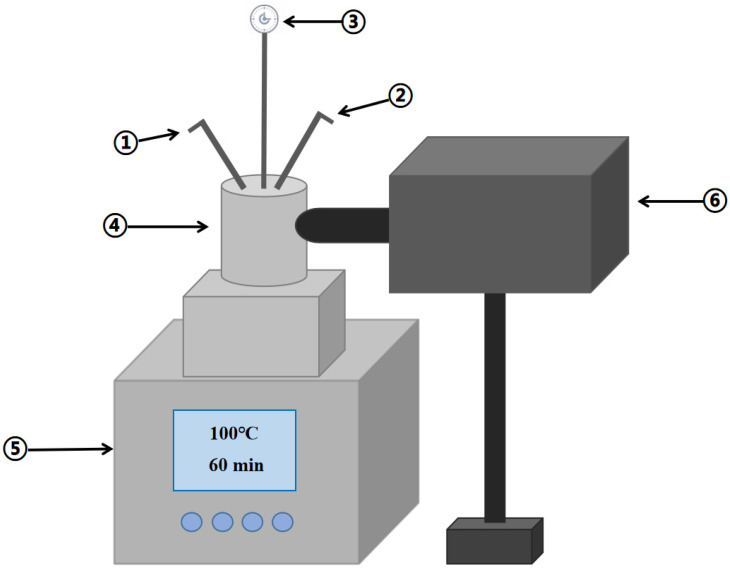
Schematic diagram of the laboratory UV-CWPO system. (① Intake valve, ② Exhaust valve, ③ Pressure gauge, ④ Photochemical high pressure reactor, ⑤ MRSC-TFM control system, ⑥ 30W UV-deuterium lamp source)**.**

**Figure 2 ijerph-18-04906-f002:**
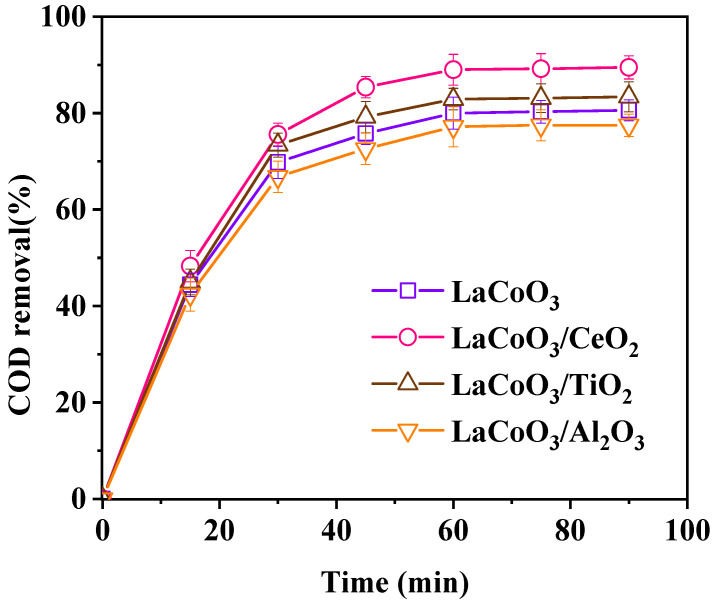
Effect of catalyst support on catalytic effect. (Catalyst preparation conditions: calcination temperature = 800 °C, calcination time = 8 h; Reaction conditions: LaCoO_3_/CeO_2_ = 0.8 g/L, H_2_O_2_ = 40 mM, 120 °C, 1 MPa, 60 min).

**Figure 3 ijerph-18-04906-f003:**
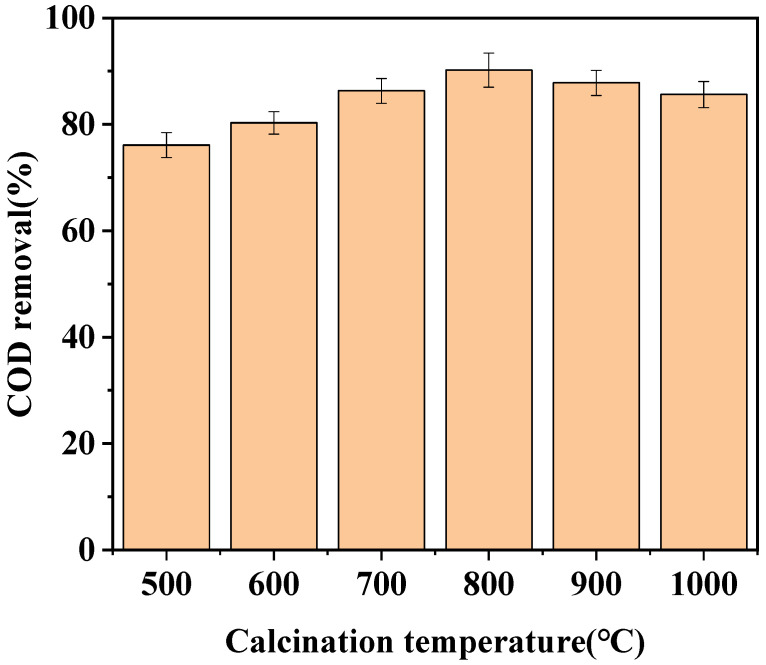
Effect of calcination temperature on catalytic effect. (Catalyst preparation conditions: catalyst support, CeO_2_, calcination time = 8 h; Reaction conditions: LaCoO_3_/CeO_2_ = 0.8 g/L, H_2_O_2_ = 40 mM, 120 °C, 1 MPa, 60 min).

**Figure 4 ijerph-18-04906-f004:**
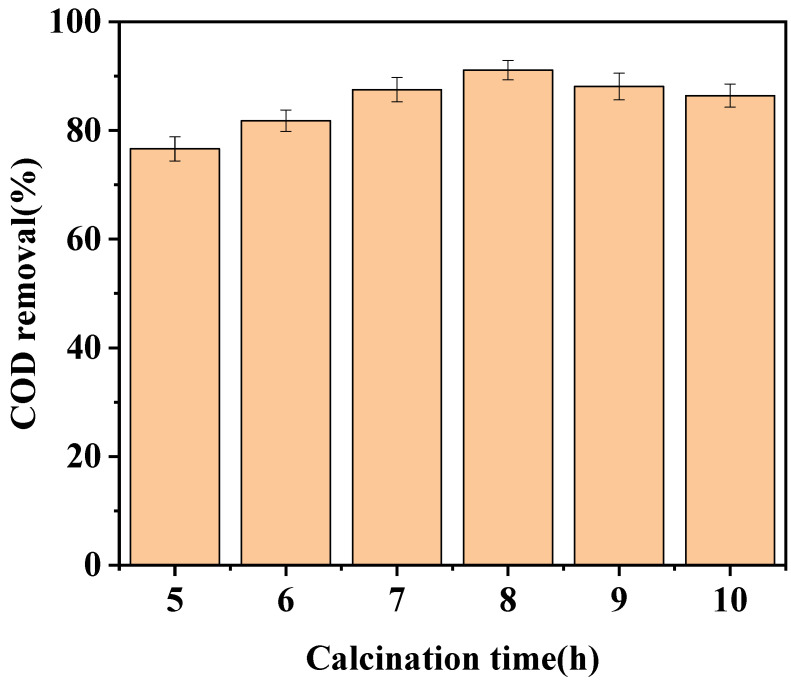
Effect of calcination time on catalytic effect. (Catalyst preparation conditions: catalyst support, CeO_2_, calcination temperature = 800 °C; Reaction conditions: LaCoO_3_/CeO_2_ = 0.8 g/L, H_2_O_2_ = 40 mM, 120 °C, 1 MPa, 60 min).

**Figure 5 ijerph-18-04906-f005:**
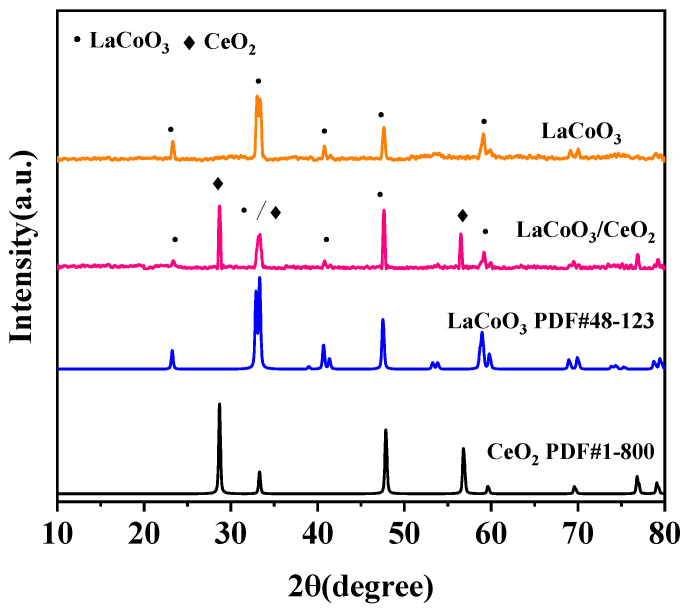
XRD patterns of LaCoO_3_ catalysts with different supports. (Catalyst preparation conditions: calcination temperature = 800 °C, calcination time = 8 h).

**Figure 6 ijerph-18-04906-f006:**
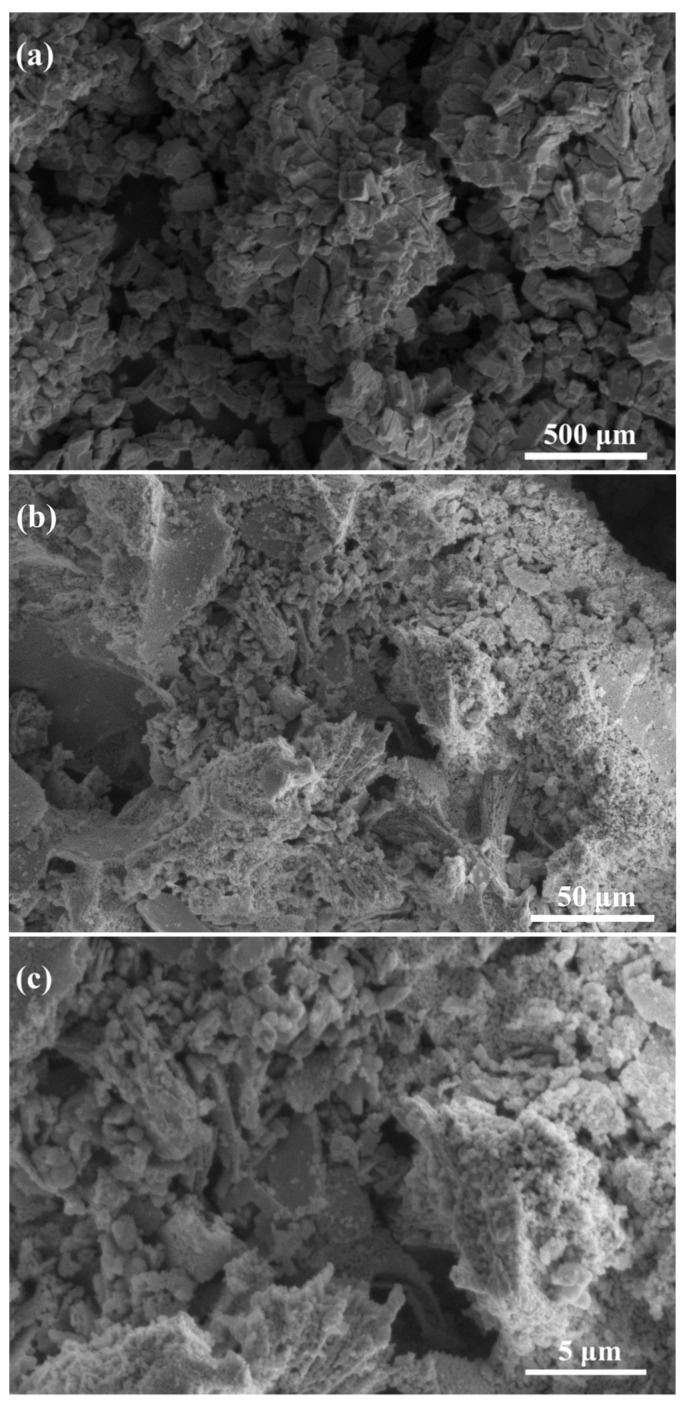
SEM images of LaCoO_3_/CeO_2_ catalyst. (**a**). SEM of 500 μm; (**b**). SEM of 50 μm; (**c**). SEM of 5 μm.

**Figure 7 ijerph-18-04906-f007:**
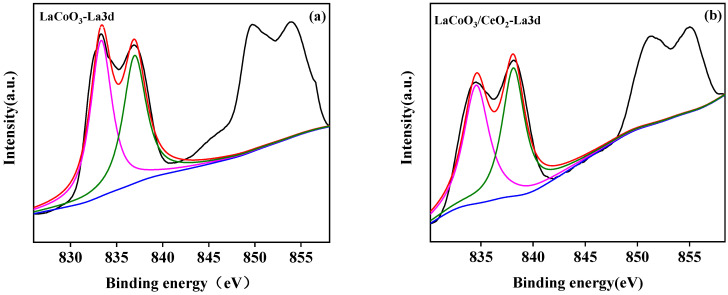
XPS spectra of La3d of LaCoO_3_ (**a**) and LaCoO_3_/CeO_2_ (**b**).

**Figure 8 ijerph-18-04906-f008:**
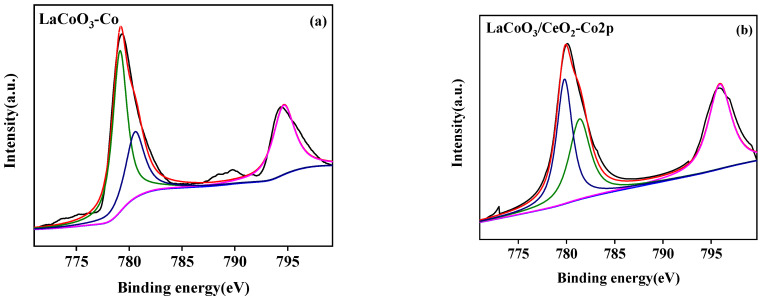
XPS spectra of Co 2p of LaCoO_3_ (**a**) and LaCoO_3_/CeO_2_ (**b**).

**Figure 9 ijerph-18-04906-f009:**
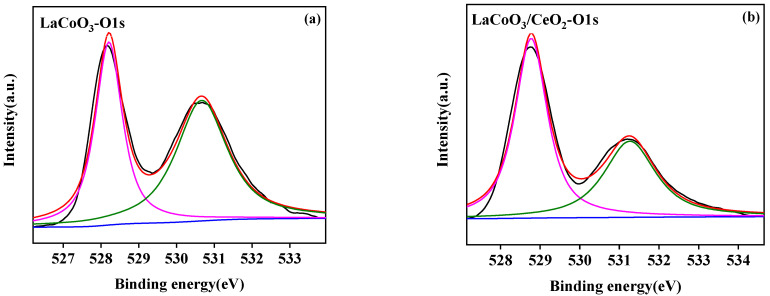
XPS spectra of O1s of LaCoO_3_ (**a**) and LaCoO_3_/CeO_2_ (**b**).

**Figure 10 ijerph-18-04906-f010:**
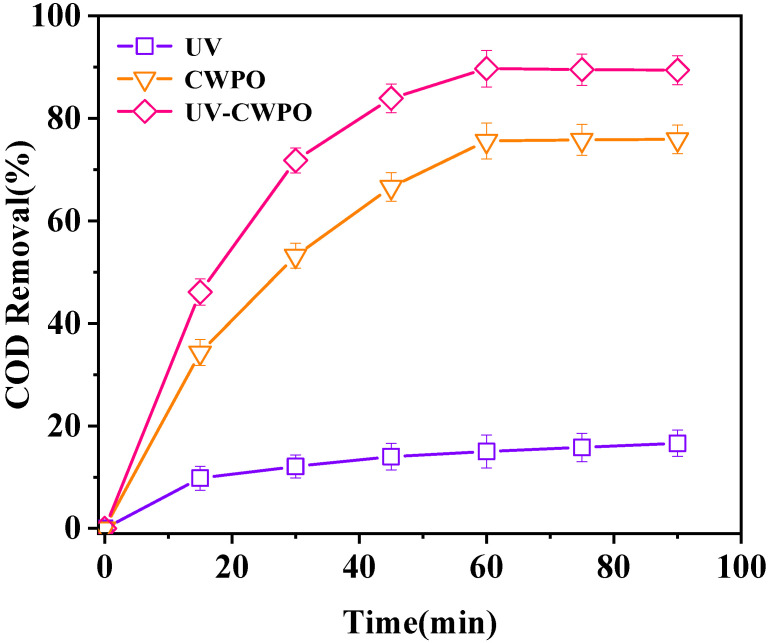
Comparison to different reaction systems. (Reaction conditions: LaCoO_3_/CeO_2_ = 1 g/L, H_2_O_2_ = 40 mM, 120 °C, 0.5 MPa).

**Figure 11 ijerph-18-04906-f011:**
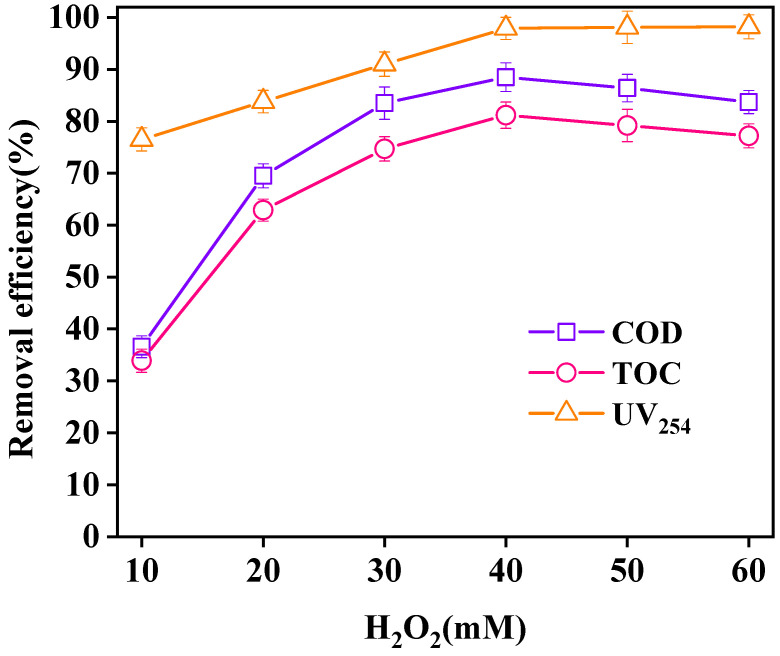
Effect of H_2_O_2_ dosage on oxidation. (Reaction conditions: LaCoO_3_/CeO_2_ = 0.8 g/L, 120 °C, 1 MPa, 60 min).

**Figure 12 ijerph-18-04906-f012:**
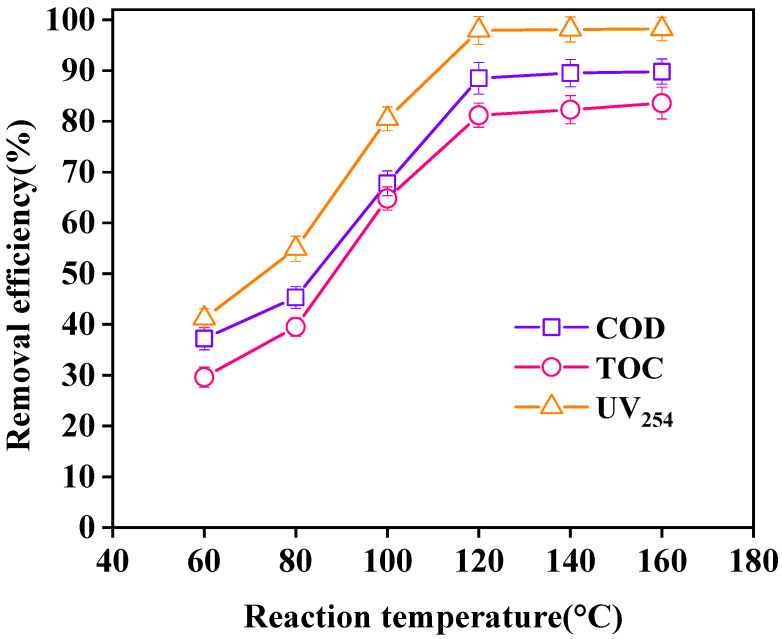
Effect of reaction temperature on oxidation. (Reaction conditions: LaCoO_3_/CeO_2_ = 0.8 g/L, H_2_O_2_ = 40 mM, 1 MPa, 60 min).

**Figure 13 ijerph-18-04906-f013:**
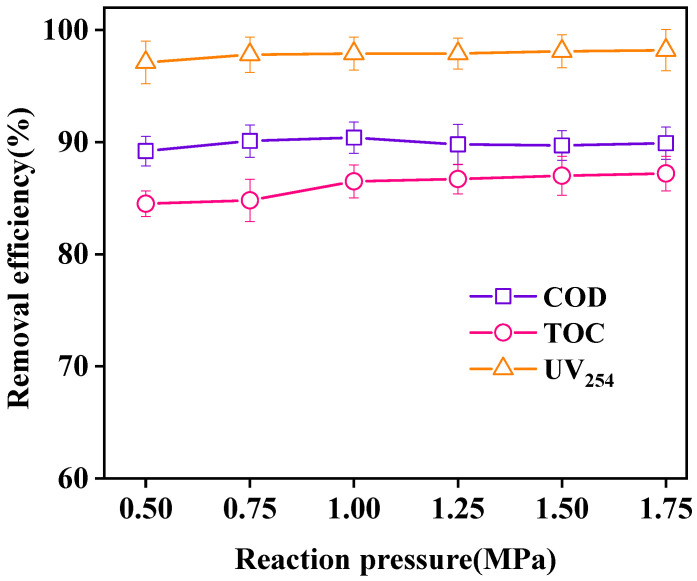
Effect of reaction pressure on oxidation. (Reaction conditions: LaCoO_3_/CeO_2_ = 0.8 g/L, H_2_O_2_ = 40 mM, 120 °C, 60 min).

**Figure 14 ijerph-18-04906-f014:**
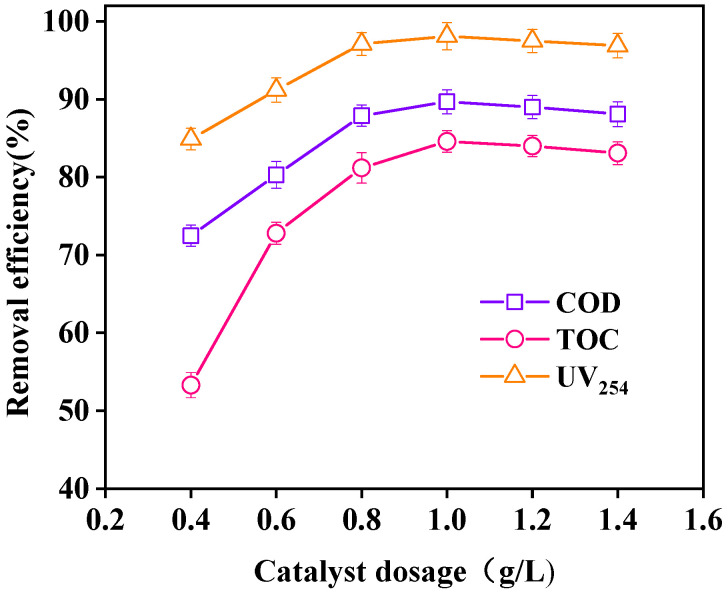
Effect of catalyst dosage on oxidation. (Reaction conditions: H_2_O_2_ = 40 mM, 120 °C, 0.5 MPa, 60 min).

**Figure 15 ijerph-18-04906-f015:**
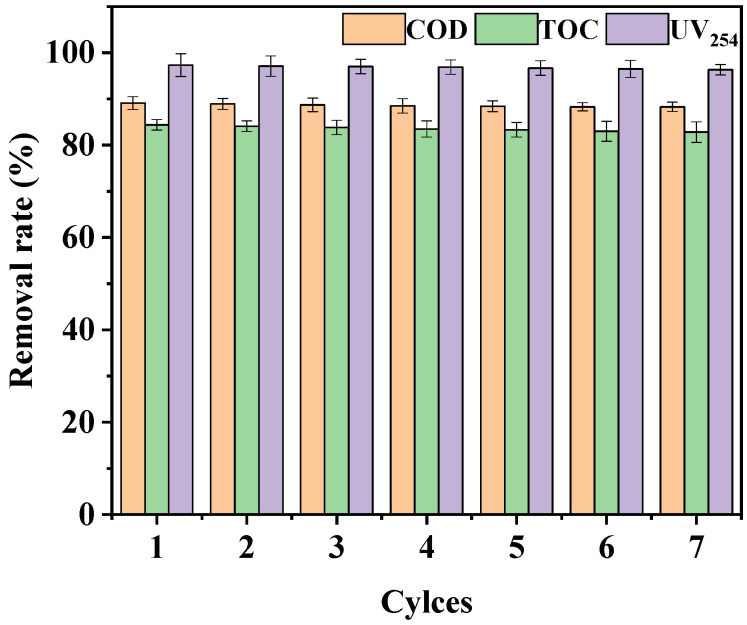
Effect of repeated use of catalyst on catalysis. (Reaction conditions: LaCoO_3_/CeO_2_ = 1 g/L, H_2_O_2_ = 40 mM, 120 °C, 0.5 MPa, 60 min).

**Figure 16 ijerph-18-04906-f016:**
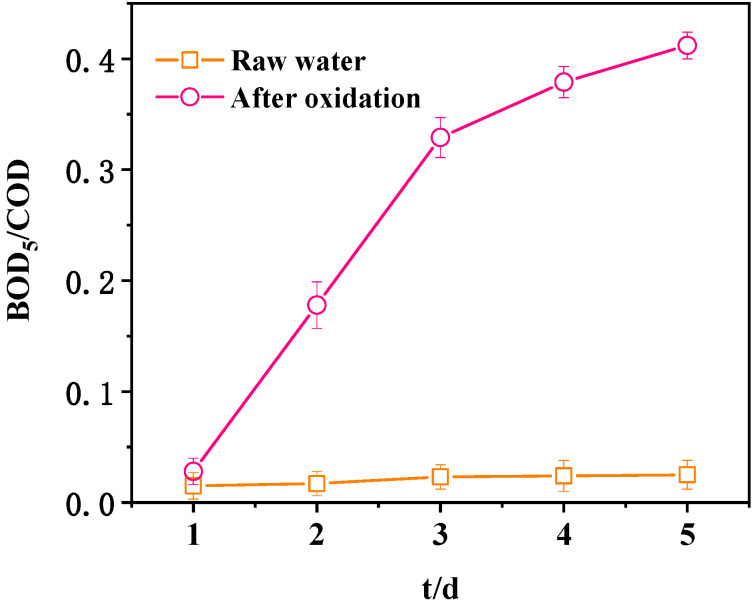
Comparison of BOD_5_/COD before and after oxidation.

**Figure 17 ijerph-18-04906-f017:**
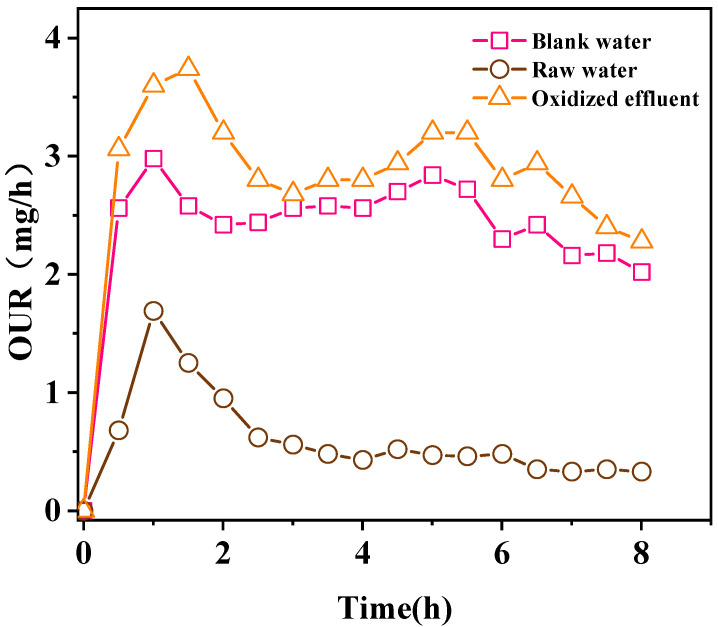
Changes in oxygen consumption rates in water samples measured and blank control samples.

**Table 1 ijerph-18-04906-t001:** BET surface area and pore structure of LaCoO_3_ catalysts with different supports.

Catalyst	Specific Surface Area (m^2^/g)	Pore Volume (cm^3^/g)	Pore Diameter (nm)
LaCoO_3_	2.6749	0.0063	4.9501
LaCoO_3_/CeO_2_	13.0729	0.0524	4.0569

**Table 2 ijerph-18-04906-t002:** Ratio of Co^3+^ and Co^2+^ and O_L_ and O_S_ of LaCoO_3_ and LaCoO_3_/CeO_2._

Catalyst	Co^3+^/Co^2+^	O_L_/O_S_
LaCoO_3_	1.88	0.80
LaCoO_3_/CeO_2_	1.15	1.48

**Table 3 ijerph-18-04906-t003:** Results of toxicity in water samples.

Sample	Respiratory Depression Rate	Toxicity
Raw water	83.66%	High toxic
Oxidation discharge	No inhibition	Non-toxic

## Data Availability

Not applicable.

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
