# Peer review of "Preparation of Supported Perovskite Catalyst to Purify Membrane Concentrate of Coal Chemical Wastewater in UV-Catalytic Wet Hydrogen Peroxide Oxidation System"

_ijerph, 2021, doi:10.3390/ijerph18094906_

Round 1

Reviewer 1 Report

The authors present a study the catalytic activity of several benchmark catalysts was tested to screen and obtain high-quality supported perovskite catalyst; to Purify Membrane Concentrate of Coal Chemical Wastewater in UV-Catalytic Wet Hydrogen Peroxide Oxidation System.

The subject is of relevant scientific interest, however, making a very general and superficial revision of the manuscript, several aspects can be observed, which are suggested to be modified.

Abstract:

Line 27 …. the ρ(BOD5)/ρ(COD) It is not known; it is suggested to describe.

Introduction

Line 73 …catalyst LaCoO3/X; The effects of calcination…., change for, …catalyst LaCoO3/X. The effects of calcination.

Content of Manuscript

Line 94 (resistivity≥1 MΩ·cm). change for, …(resistivity ≥1 MΩ·cm)

Line 100-101 According to the stoichiometric ratio N(La(NO3)3·6H2O) : N change for, …(La(NO3)3·6H2O): (Co(NO3)2•6H2O) =1:1,

Line 101  N (metal ion) : N (C6H8O7•H2O) = 1:1.5, change for, … (metal ion):(C6H8O7•H2O) = 1:1.5

Line 106     105 °C until they change for, …105°C until they

Line 108   temperature(500 °C–1000°C) for the corresponding time(5h–10h). change for, … temperature (500°C–1000°C) for the corresponding time (5h–10h).

Line 135-136 UV-Vis spectrophotometer (UV-2600, Shimadzu Company, Japan) was used to measure water   UV254 at 254 nm wavelength. Is necessary to indique why at 254 nm wavelength.

Line 153 3.1.1. Effect of Catalyst Support. In this section is necessary to specific the calcination temperature

Line 174 3.1.2. Effect of Calcination Temperature. In this section is necessary to specific the calcination time

Line 185 -catalytic activity[19].   change for, catalytic activity [19].

Line 188 effect[20]. change for, effect [20].

Line 192 3.1.3. Effect of Calcination Time.  In this section is necessary to specific use 800°C of temperature

Line 212 3.2.1. X-ray diffraction (XRD). In this section is necessary to specific that catalysts calcination temperature was 800°C and time 8 h. Is the same treatment for perovskite without supported?

Line 226 3.2.2. Brunauer-Emmett-Teller (BET).  Suggested to change for: N2 adsorption/desorption analysis

In this section is necessary to improve

Line 227 The specific surface area and pore structure information. Suggested to change “pore structure” for “pore texture”

Line 229    the pore volume increased from 0.006297 cm3/g to 0.052408 cm3/g. Suggested to change for from 0.0063 cm3/g to 0.0524 cm3/g

Line 234 3.2.3. Scanning Electron Microscopy (SEM)

In this section it is necessary to review the description of the following two terms

-indicates a good state of surface dispersion.

-evenly distributed pores

By SEM it is not possible to determine either the surface dispersion, nor the uniform distribution of the pores

Line 255    were attributed to La3d 3/2 and La3d 5/2. Suggested to change for La 3d3/2 and La 3d5/2

Line 273-277 It is necessary to review the follows affirmations and it is suggested improve the text

Oxygen vacancy was formed in the charge balance process, which was conducive to the improvement of the catalytic activity [29]. The ratio of OL/OS increased from 0.80 to 1.48, indicating that the number of lattice oxygen defects gradually increased in the catalyst after loading, thereby increasing the active point of the catalyst and improving the catalytic effect [30].

Line 286 3.3.1. Effect of H2O2 Dosage on Wastewater Purification. In this section is necessary to specific: reaction temperature, reaction pressure and catalyst dosage employed.

Line 292-294 It is necessary to review the follow affirmation

With increasing H2O2 dosage, the yield of •OH in the system continued to increase, which enhanced the oxidation capacity of the reaction system.

Line 300 3.3.2. Effect of Reaction Temperature on Wastewater Purification. In this section is necessary to specific H2O2 dosage, reaction pressure and catalyst dosage employed.

Line 313 3.3.3. Effect of Reaction Pressure on Effluent Purification. In this section is necessary to specific H2O2 dosage, reaction temperature and catalyst dosage employed.

Line 327 3.3.4. Effect of Catalyst Dosage on Wastewater Purification. In this section is necessary to specific H2O2 dosage, reaction temperature and reaction pressure.

Line 340 3.4. Reusability and Stability of Catalyst.  In this section is necessary to specific H2O2 dosage, reaction temperature, reaction pressure and catalyst dosage employed.

Line 348 Biodegradability ANALYSIs of Oxidized Effluent. Suggested to change for Biodegradability Analysis

It is necessary to review and it is suggested improve the redaction

Line 364 4.2. Biological Toxicity Analysis

In this section is necessary to review and it is suggested improve the redaction

Line 386 5. Conclusions. In this section is necessary to review and it is suggested improve the redaction

Although, the document contains some interesting elements, I think it should be thoroughly reviewed and corrected.

Author Response

Response to Reviewer 1 Comments

Dear reviewers:

Thank you very much for your careful review and constructive suggestions with regard to our manuscript “Preparation of supported perovskite catalyst to purify membrane concentrate of coal chemical wastewater in UV-catalytic wet hydrogen peroxide oxidation system” (ID: ijerph-1165573). The comment is helpful for authors to revise and improve our paper. We have revised the manuscript and made changes in the manuscript according to the referees′ good comments. We appreciate for Editors/Reviewers’ warm work earnestly, and hope that the corrections will meet with approval.

The following is our response to the comments and suggestions. All the changes are marked in red in the revised manuscript.

Point 1: Line 27 …. the ρ(BOD5)/ρ(COD) It is not known; it is suggested to describe.

Response 1: Thank you for the comments on this paper. The ρ(BOD5)/ρ(COD) referred to the ratio of BOD5 to COD. For ease of understanding, ρ(BOD5)/ρ(COD) has been changed to BOD5/COD.

Under the optimal operating conditions, the oxidized effluent changed from high toxicity to non-toxicity, the BOD5/COD increased from 0.02 to 0.412, and the biodegradability of the oxidized effluent was greatly improved.

First, BOD5/COD before and after wastewater purification in the UV-CWPO system was tested (Figure 16).

The BOD5/COD was only 0.02 after 5 days of biochemical cultivation, indicating that wastewater before oxidation treatment could not be directly treated in the biochemical unit.

The BOD5/COD of the wastewater reached 0.412 after 5 days of biochemical culture, and the biodegradability was greatly improved.

 Please see the information above in Page 1, line 26~28; Page 14, line 399~400; Page 14~15, line 402~406.

Point 2:

-Line 73 …catalyst LaCoO3/X; The effects of calcination…., change for, …catalyst LaCoO3/X. The effects of calcination.

-Line 94 (resistivity≥1 MΩ·cm). change for, …(resistivity ≥1 MΩ·cm)

-Line 100-101 According to the stoichiometric ratio N(La(NO3)3·6H2O) : N change for, …(La(NO3)3·6H2O): (Co(NO3)2·6H2O) =1:1,

-Line 101  N (metal ion) : N (C6H8O7·H2O) = 1:1.5, change for, … (metal ion):(C6H8O7·H2O) = 1:1.5

-Line 106  105 °C until they change for, …105°C until they

-Line 108   temperature(500 °C–1000°C) for the corresponding time(5h–10h). change for, …temperature (500°C–1000°C) for the corresponding time (5h–10h).

Response 2: Thank you for the comments on this paper. According to your suggestions, we have checked the punctuation marks and spaces in our manuscript, and revised them one by one.

The catalytic activity of several benchmark catalysts was tested to screen a desirable carrier and obtain high-quality supported perovskite catalyst LaCoO3/X. The effects of calcination temperature and duration on the wastewater purification efficiency of the catalyst were investigated by batch experiments.

Deionized water was produced by a pure water treatment system (EPED-Z1-30T, Eped Technology, China) and used to prepare all solutions (resistivity ≥1 MΩ·cm).

According to the stoichiometric ratio (La(NO3)3·6H2O): (Co(NO3)2·6H2O)=1:1, (metal ion):(C6H8O7·H2O)=1:1.5, certain amounts of La(NO3)3·6H2O, Co(NO3)2·6H2O, and C6H8O7·H2O were weighed and dissolved in deionized water to form a mixed solution.

The samples were dried at 105°C until they were completely dried. The samples were finely ground and placed in a muffle furnace.

 Perovskite catalyst LaCoO3/X was obtained by roasting at the corresponding temperature(500°C–1000°C) for the corresponding time(5h–10h).

 Please see the information above in Page 2, line 79~81; Page 3, line 100~102; Page 3, line 109~112; Page 3, line 114~117.

Point 3: Line 135-136 UV-Vis spectrophotometer (UV-2600, Shimadzu Company, Japan) was used to measure water UV254 at 254 nm wavelength. Is necessary to indique why at 254 nm wavelength.

Response 3: Thank you for the comments on this paper. The UV254 value is the absorbance of some organic matter in water under the ultraviolet light at 254nm. UV254 reflects the quantity of humus macromolecular organic compounds and aromatic compounds containing C=C double bond and C=O double bond naturally occurring in water. Phenolic compounds were the main representative compounds in the membrane concentrate, thereby it was necessary to chose UV254 as the water quality index. We have provided additional information in Section 2.3.

Generally speaking, the UV254 value is the absorbance of some organic matter in water under the ultraviolet light at 254nm. UV254 reflects the quantity of humus macromolecular organic compounds and aromatic compounds containing C=C double bond and C=O double bond naturally occurring in water. Phenolic compounds were the main representative compounds in the membrane concentrate, thereby it was necessary to chose UV254 as the water quality index.

 Please see the information above in Page 4, line 148~154.

Point 4:

-Line 153 3.1.1. Effect of Catalyst Support. In this section is necessary to specific the calcination temperature

-Line 174 3.1.2. Effect of Calcination Temperature. In this section is necessary to specific the calcination time

-Line 192 3.1.3. Effect of Calcination Time.  In this section is necessary to specific use 800°C of temperature

Response 4: Thank you for the comments on this paper. We have attached the catalyst preparation conditions after the Figure names of Figure 2, Figure 3 and Figure 4. In addition, the experimental conditions for verifying the effect of the catalyst have also been annotated.

 Please see the information above in Page 5, line 197~199; Page 6, line 217~219; Page 7, line 237~239.

Point 5:

-Line 185 -catalytic activity[19].   change for, catalytic activity [19].

-Line 188 effect[20]. change for, effect [20].

Response 5: Thank you for the comments on this paper. We have checked the symbols in the full manuscript and modified the improper format one by one.

The perovskite structure of the catalyst on the CeO2 carrier had not been formed when the calcination temperature was in a lower range, and the purity of the effective perovskite structure was also relatively low, resulting in lower catalyst activity [20].

 Meanwhile, if the calcination temperature was too high, the catalyst was prone to sintering, resulting in the collapse of the internal structure of the catalyst, blockage of the pores, and reduction of active sites, which in turn leaded to the degradation of the catalytic activity [21].

 Please see the information above in Page 5~6, line 208~214.

Point 6: Line 212 3.2.1. X-ray diffraction (XRD). In this section is necessary to specific that catalysts calcination temperature was 800°C and time 8 h. Is the same treatment for perovskite without supported?

Response 6: Thank you for the comments on this paper. Perovskite without supported was also synthesized under the same conditions (calcination temperature = 800°C, calcination time = 8 h). The preparation conditions were attached to the name of the Figure in Figure 5.

 Please see the information above in Page 7~8, line 254 ~256.

Point 7:

-Line 226 3.2.2. Brunauer-Emmett-Teller (BET).  Suggested to change for: N2 adsorption/desorption analysis

-Line 227 The specific surface area and pore structure information. Suggested to change “pore structure” for “pore texture”

-Line 229    the pore volume increased from 0.006297 cm3/g to 0.052408 cm3/g. Suggested to change for from 0.0063 cm3/g to 0.0524 cm3/g

-Line 255    were attributed to La3d 3/2 and La3d 5/2. Suggested to change for La 3d3/2 and La 3d5/2

Response 7: Thank you for the comments on this paper. We have changed the inappropriate noun representation and have unified the number of decimal places.

3.2.2. N2 Adsorption/Desorption Analysis

The specific surface area and pore texture information of LaCoO3 and LaCoO3/CeO2 are listed in Table 1.

Catalyst

Specific Surface Area (m2/g)

Pore Volume (cm3/g)

Pore Diameter (nm)

LaCoO3

2.6749

0.0063

4.9501

LaCoO3/CeO2

13.0729

0.0524

4.0569

Two pairs of bimodals located at the lower binding energy (A: 833.8 and 837.7 eV; B: 834.2 and 838.1 eV) were attributed to La 3d3/2 and La 3d5/2.

 Please see the information above in Page 8, line 256~258; Page 8, line 263; Page 9, line 284~285.

Point 8:Line 234 3.2.3. Scanning Electron Microscopy (SEM)

In this section it is necessary to review the description of the following two terms

-indicates a good state of surface dispersion.

-evenly distributed pores

By SEM it is not possible to determine either the surface dispersion, nor the uniform distribution of the pores

Response 8: Thank you for the comments on this paper. Your suggestion was very accurate. SEM cannot determine surface dispersion and uniform distribution of pores, so the language described here has been modified. The electron microscope of this paper shows that there are many pores.

Although particles were closely connected, the gap between them were clearly seen. In addition, although the LaCoO3 particles on the surface of the support are partially sintered, there are still a large number of pores on the crystal surface and between the crystal structures (Fig.5 (b) and 5(c)).

 Please see the information above in Page 8, line 267~271.

Point 9:Line 273-277 It is necessary to review the follows affirmations and it is suggested improve the text

Oxygen vacancy was formed in the charge balance process, which was conducive to the improvement of the catalytic activity [29]. The ratio of OL/OS increased from 0.80 to 1.48, indicating that the number of lattice oxygen defects gradually increased in the catalyst after loading, thereby increasing the active point of the catalyst and improving the catalytic effect [30].

Response 9: Thank you for the comments on this paper. We have reviewed the affirmations and improved the manuscript.

The oxygen vacancy was formed during charge balance, which was beneficial to the improvement of catalytic activity [30]. The OL/OS ratio increased from 0.80 to 1.48, indicating that the lattice oxygen defects increased in the supported catalysts. Lattice oxygen defect was beneficial to increase the active point of the catalyst, thus improving the catalytic effect of the catalyst [31].

 Please see the information above in Page 9, line 304~308.

Point 10:

-Line 286 3.3.1. Effect of H2O2 Dosage on Wastewater Purification. In this section is necessary to specific: reaction temperature, reaction pressure and catalyst dosage employed.

-Line 300 3.3.2. Effect of Reaction Temperature on Wastewater Purification. In this section is necessary to specific H2O2 dosage, reaction pressure and catalyst dosage employed.

-Line 313 3.3.3. Effect of Reaction Pressure on Effluent Purification. In this section is necessary to specific H2O2 dosage, reaction temperature and catalyst dosage employed.

-Line 327 3.3.4. Effect of Catalyst Dosage on Wastewater Purification. In this section is necessary to specific H2Odosage, reaction temperature and reaction pressure.

-Line 340 3.4. Reusability and Stability of Catalyst.  In this section is necessary to specific H2O2 dosage, reaction temperature, reaction pressure and catalyst dosage employed.

Response 10: Thank you for the comments on this paper. We have written the reaction conditions after the Figure names in Figure 11, Figure 12, Figure 13, Figure 14, and Figure 15.

 Please see the information above in Page 12, line 342~343; Page 12, line 356~357; Page 13, line 371~372; Page 14, line 385~386.

Point 11:Line 292-294 It is necessary to review the follow affirmation

With increasing H2O2 dosage, the yield of •OH in the system continued to increase, which enhanced the oxidation capacity of the reaction system.

Response 11: Thank you for the comments on this paper. We have reviewed the affirmations and improved the manuscript.

With the increase of H2O2 dosage, more ·OH was produced, which enhanced the oxidation capacity of the reaction system [31].

 Please see the information above in Page 11, line 336~337.

Point 12:Line 348 Biodegradability ANALYSIs of Oxidized Effluent. Suggested to change for Biodegradability Analysis

Response 12: Thank you for the comments on this paper.  Biodegradability Analysis of Oxidized Effluent has been replaced by Biodegradability Analysis.

  1. Biodegradability Analysis

 Please see the information above in Page 14, line 396.

Point 13:

-It is necessary to review and it is suggested improve the redaction

Line 364 4.2. Biological Toxicity Analysis

-In this section is necessary to review and it is suggested improve the redaction

Line 386 5. Conclusions. In this section is necessary to review and it is suggested improve the redaction

Response 13: Thank you for the comments on this paper. We have reviewed the format of the full text and revise it one by one.

4.2. Biological Toxicity Analysis

In this study, supported perovskite catalyst LaCoO3/X was prepared by two steps (sol–gel method and impregnation).

(calcination temperature of 800°C, calcination time of 8 h).

The overall crystal structure of LaCoO3/CeO2 was clear, and the composite metal oxide LaCoO3 with standard perovskite structure was well distributed on the surface of the CeO2 carrier. LaCoO3/CeO2 was used as the catalyst in the UV-CWPO system to participate in the purification of coal chemical wastewater membrane concentrate. Under the best process conditions, i.e., pH of 3, reaction time of 60 min, H2O2 dosage of 40 mM, reaction temperature of 120°C, reaction pressure of 0.5 MPa and catalyst dosage of 1 g/L, the removal efficiency of COD, TOC, and UV254 reached 89.7%, 84.6%, and 98.1%, respectively.

Apparently, the prepared LaCoO3/CeO2 catalyst showed desirable reusability and stability. 

The biodegradability analysis showed that the BOD5/COD of wastewater treated by the UV-CWPO system increased from 0.02 to 0.412.

The toxicity analysis indicated that the water sample after oxidation changed from the original highly toxic to non-toxic, and the oxidized effluent was suitable for subsequent biochemical unit. Hence, the supported perovskite catalyst LaCoO3/CeO2 is a pure perovskite catalyst with orthogonal crystal form and has a good application prospect to treating concentrated liquid of coal chemical wastewater membrane in UV-CWPO system.

 Please see the information above in Page 15, line 413; Page 16, line 436~437; Page 16, line 439~440; Page 16, line 440~447; Page 16, line 449~450; Page 16, line 451~452; Page 16, line 452~457; .

Reviewer 2 Report

The article details the development of a catalyst for the purification of membrane concentrates from Coal Chemical Wastewater using a great variety of techniques for its characterization as well as a search for the best operating conditions.

The article is very interesting and well structured, however the writing needs to be significantly improved. The English is very poor and the sentences are very redundant. In one paragraph the same thing is repeated several times in different sentences, which makes the reading very heavy. It is a pity, as it seriously affects the quality of the work done. 

Furthermore, other issues would be addressed before publication:

-From which membrane process is the effluent coming?

-An scheme of the reactor would be included for a better understanding

-Manuscript talk about a photocatalytic system, however it does not give details about which was the light source or the power of the lamp. Moreover, the influence of the light source in the process was not reported or evaluated, there are no blanks either. Have light any role in the degradation process?

-Regarding concentrate depuration, in every section is reported the evaluation of one of the parameters of the process, but the rest of the conditions of such experiments were not described in the text. It should be explained for a better understanding.

-H2O2 concentration is given in mL/L. For a better understanding and to be able to compare with other treatments and process it should be in mg/L.

-Abbreviations are not properly used. OUR is not described in the text and then, instead of describing it as Oxygen Uptake Rate is described as "aerobic rate" in lines 368 369 and 371 and also as oxygen consumption rate in line 370. This should be corrected.

-line 96: "The sample had..." better to say "Membrane concentrate characteristics were..."

-line 143: "judge" should be change to "assess" or "evaluate"

-section 3.1.1: in all cases COD removal reaches a plateau at 60 min thus it would be better to remark that, instead of 90 min, as it is a lower treatment time.

-line 180: "Continuously increase" it seems that it talks about a temperature ramp. Better to avoid "continuously"

-line 200 is the same as line 198 (just a very evident example of the several redundancies found in the text)

-•OH: OH, the dot in superscrip

-line 341: There is not a lose in efficiency and then a process to recover it thus the manuscript does not report the "recoverability", it evaluates stability or duration.

-line 345: There is not an economical assessment, thus that term should be avoided.

-lines 348 and 349: review the format of the titles

-section 4: how much was the biodegradability improved ? which is the reference value? usually is 0.2 but it is not reported.

-line 376: "can be better utilized" microorganisms does not utilize the water, this term should be change by other such as "easily degrade".

Author Response

Please check in the attachment.

Round 2

Reviewer 2 Report

The quality of the manuscript has been significantly improved after revision. However some minor changes should be addressed before publication.

Line 52. space before "Zazo et al."

Line 146. "Generally speaking" is in italic, is it correct?

Line 148. "Humus" should be changed by "humic"

Lines 300-303 are in other character type.

Line 319. The text says that there is a synergistic effect of UV and CWPO, however in the results COD removal by UV=16.6%; CWPO=75.6% and UV-CWPO=89.7%, lower than the sum of UV and CWPO (UV+CWPO=92.2%), thus UV improves CWPO but there were not synergistic effect.

Lines 396-400: Biodegradability explanation is correct. However, manuscript says that  "BOD5/COD was only 0.02" but does not give the value to consider an effluent biodegradable. For someone which is not familiar with this kind of analysis/term including the value at which an effluent is considered biodegradable facilitates the understanding of the text.

Figures. In all is missing the space between ":" and the next word, and the space between the number ans its units

Author Response

Response to Reviewer 2 Comments

Point 1:

-Line 52. space before "Zazo et al."

-Line 146. "Generally speaking" is in italic, is it correct?

-Line 148. "Humus" should be changed by "humic"

-Lines 300-303 are in other character type.

Response 1: Thank you for the comments on this paper.  According to your suggestions, we have checked the inappropriate statements and formatting errors in our manuscript, and revised them one by one.

Some studies have shown the superiority of photocatalytic system. Zazo et al. [7] found that UV radiation can promote the reduction of Fe3+ to Fe2+ on the surface of the catalyst, so that more H2O2 can be decomposed into hydroxyl radical ·OH.

Generally speaking, the UV254 value is the absorbance of some organic matter in water under the ultraviolet light at 254nm.

UV254 reflects the quantity of humic macromolecular organic compounds and aromatic compounds containing C=C double bond and C=O double bond naturally occurring in water.

The oxygen vacancy was formed during charge balance, which was beneficial to the improvement of catalytic activity [30]. The OL/OS ratio increased from 0.80 to 1.48, indicating that the lattice oxygen defects increased in the supported catalysts. Lattice oxygen defect was beneficial to increase the active point of the catalyst, thus improving the catalytic effect of the catalyst [31].

Please see the information above in Page 2, line 53~55; Page 4, line 148~150;  Page 4, line 150~152; Page 9, line 304~308.

Point 2: Line 319. The text says that there is a synergistic effect of UV and CWPO, however in the results COD removal by UV=16.6%; CWPO=75.6% and UV-CWPO=89.7%, lower than the sum of UV and CWPO (UV+CWPO=92.2%), thus UV improves CWPO but there were not synergistic effect.

Response 2: Thank you for the comments on this paper. We have reviewed the affirmations and improved the manuscript.

Therefore, the UV-CWPO system proves that the combination of UV and CWPO technologies can produce beneficial effects and improve the COD removal efficiency.

Please see the information above in Page 11, line 324~325.

Point 3: Lines 396-400: Biodegradability explanation is correct. However, manuscript says that  "BOD5/COD was only 0.02" but does not give the value to consider an effluent biodegradable. For someone which is not familiar with this kind of analysis/term including the value at which an effluent is considered biodegradable facilitates the understanding of the text.

Response 3: Thank you for the comments on this paper. We have improved the manuscript according to your suggestion.

In general, the greater the BOD5/COD value is, the better the biodegradability of wastewater is. The wastewater is generally considered to be biodegradable when BOD5/COD>0.3. The biodegradability of the concentrated liquid of coal chemical wastewater membrane before oxidation was very poor. The BOD5/COD was only 0.02 after 5 days of biochemical cultivation, indicating that wastewater before oxidation treatment could not be directly treated in the biochemical unit. The biodegradability of wastewater purified by this system was greatly improved. The BOD5/COD of the wastewater reached 0.412 after 5 days of biochemical culture, and the biodegradability was greatly improved.

Please see the information above in Page 14, line 401~403.

Point 4: Figures. In all is missing the space between ":" and the next word, and the space between the number ans its units

Response 4: Thank you for the comments on this paper. According to your suggestions, we have checked the punctuation marks and spaces in our manuscript, and revised them one by one.

Figure 2. Effect of catalyst support on catalytic effect. (Catalyst preparation conditions: calcination temperature = 800°C, calcination time = 8 h; Reaction conditions: LaCoO3/CeO2 = 0.8 g/L, H2O2 = 40 mM, 120°C, 1 MPa, 60 min)

Figure 3. Effect of calcination temperature on catalytic effect. (Catalyst preparation conditions: catalyst support, CeO2, calcination time = 8 h; Reaction conditions: LaCoO3/CeO2 = 0.8 g/L, H2O2 = 40 mM, 120°C, 1 MPa, 60 min)

Figure 4. Effect of calcination time on catalytic effect. (Catalyst preparation conditions: catalyst support, CeO2, calcination temperature = 800°C; Reaction conditions: LaCoO3/CeO2 = 0.8 g/L, H2O2 = 40 mM, 120°C, 1 MPa, 60 min)

Figure 5. XRD patterns of LaCoO3 catalysts with different supports. (Catalyst preparation conditions: calcination temperature = 800°C , calcination time = 8 h)

Figure 10. Comparison to different reaction systems. (Reaction conditions: LaCoO3/CeO2 = 1 g/L, H2O2 = 40 mM, 120°C, 0.5 MPa)

Figure 11. Effect of H2O2 dosage on oxidation. (Reaction conditions: LaCoO3/CeO2 = 0.8 g/L, 120°C, 1 MPa, 60 min)

Figure 12. Effect of reaction temperature on oxidation. (Reaction conditions: LaCoO3/CeO2 = 0.8 g/L, H2O2 = 40 mM, 1 MPa, 60 min)

Figure 13. Effect of reaction pressure on oxidation.(Reaction conditions: LaCoO3/CeO2 = 0.8 g/L, H2O2 = 40 mM, 120°C, 60 min)

Figure 14. Effect of catalyst dosage on oxidation. (Reaction conditions: H2O2 = 40 mM, 120°C, 0.5 MPa, 60 min)

Figure 15. Effect of repeated use of catalyst on catalysis. (Reaction conditions: LaCoO3/CeO2 = 1 g/L, H2O2 = 40 mM, 120°C, 0.5 MPa, 60 min)

Please see the information above in Page 5, line 197~199; Page 6, line 217~219;  Page 7, line 237~239; Page 7, line 254~255; Page 11, line 327~328;  Page 12, line 342~343; Page 12, line 356~357; Page 13, line 371~372;  Page 14, line 386~387; Page 14, line395~396.